

# Annual plankton community metabolism in estuarine and coastal waters in Perth (Western Australia)

Susana Agusti[1,*], Lorena Vigoya[2,3,*] and Carlos Manuel Duarte[1,*]

[1] Red Sea Research Center, King Abdullah University of Science and Technology (KAUST), Thuwal, Saudi Arabia
[2] The UWA Oceans Institute, University of Western Australia, Crawley, WA, Australia
[3] Present address: AECOM, Auckland, New Zealand
[*] These authors contributed equally to this work.

Corresponding author
Susana Agusti,
susana.agusti@kaust.edu.sa

## ABSTRACT

The planktonic metabolic balance that is the balance between gross primary production (GPP) and community respiration (CR) was determined in Matilda Bay (estuarine) and Woodman Point (coastal) in Perth, Western Australia. The rates of net community production (NCP = GPP – CR) and the ratio between GPP and CR (P/R) were assessed to evaluate whether the metabolic balance in the two coastal locations tends to be net autotrophic (production exceeding community respiration) or net heterotrophic (respiration exceeding production). We also analyzed environmental variability by measuring temperature, salinity, and nutrients and chlorophyll $a$ concentration. Samples were collected biweekly from March 2014 to March 2015. During the study period the metabolic rates were three times higher in Matilda Bay than in Woodman Point. The predominant metabolism was net autotrophic at both sites with P/R ratios >1 in the majority of the sampling dates. In Matilda Bay, the metabolic rates were negatively correlated with salinity denoting river dynamics influence, and positively with chlorophyll $a$. In Woodman Point only the GPP was positively correlated with chlorophyll $a$. The positive correlation between P/R ratio and GPP in Matilda Bay and the positive correlations between the metabolic rates and chlorophyll $a$ suggest that factors controlling autotrophic processes are modulating the planktonic metabolic balance in the coastal marine ecosystem in Perth. Significant correlations were found between CR and GPP-standardized to chlorophyll $a$ and water temperature. The net autotrophic metabolic balance indicates that in both ecosystems planktonic communities are acting as a sink of $CO_2$ and as a source of organic matter and oxygen to the system and are able to export organic matter to other ecosystems.

Subjects Ecology, Marine Biology, Biogeochemistry, Biological Oceanography
Keywords Planktonic metabolic balance, Net community production (NCP), Gross primary production (GPP), Community respiration (CR), Indian Ocean, Swan river estuary, Temperature

## INTRODUCTION

Plankton metabolism is a fundamental property of marine ecosystem driving the flux of gases and the transference of organic matter to the food web (*Duarte, Agustí & Regaudie-de Gioux, 2011*). The metabolism of plankton communities in the open ocean is in approximate balance, i.e., with gross primary production (GPP) similar to community
respiration (CR) and a P/R ratio close to 1.0, or experiences small deviations from this balance (*Williams et al., 2013*; *Duarte et al., 2013*), because deviations from such balance require external fluxes of nutrients or organic carbon, which are small. In contrast, coastal plankton communities, which typically present higher metabolic rates (*Duarte & Agustí, 1998*), may have large deviations from metabolic balance with either excess respiration over production when the ecosystem receives large inputs of labile organic carbon (e.g., Mediterranean coastal areas, *Duarte, Agustí & Vaqué, 2004*; *Vidussi et al., 2011*) or gross primary production in excess of respiration when the ecosystem receives large inputs of dissolved inorganic nutrients (e.g., *Agustí, Satta & Mura, 2004*).

The metabolic balance of coastal ecosystems plays an important role in determining their role as $CO_2$ sources or sinks (*Borges, 2005*; *Cai, 2011*). Recently, a contrasting role between continental shelves acting as sinks and near-shore ecosystems as sources of atmospheric $CO_2$ was proposed to reconcile opposing views on the role of coastal ecosystems as $CO_2$ sources or sinks (*Chen & Borges, 2009*; *Cai, 2011*). In particular, inner estuaries are believed to act as sources of $CO_2$ to the atmosphere due to a prevalence of heterotrophic ecosystem metabolic status fueled by land-derived inputs of organic carbon whereas outer reaches of estuaries tend to be $CO_2$ sinks (*Odum & Hoskin, 1958*; *Odum & Wilson, 1962*; *Heip et al., 1995*; *Kemp et al., 1997*; *Gattuso, Frankignoulle & Wollast, 1998*; *Hopkinson & Smith, 2005*).

However, virtually all of the results from near-shore and open coastal ecosystems thus far refer to those in the northern hemisphere, particularly Europe, the USA and Asia (*Borges, 2005*; *Chen & Borges, 2009*; *Cai, 2011*). As these typically represent highly populated areas with watersheds supporting intense agricultural practices, the results may not be directly transferable to coastal areas in the southern hemisphere. The metabolism of Australian coastal waters was recently studied by *McKinnon et al. (2013)* and *McKinnon et al. (2017)*. Autotrophic plankton metabolism prevailed in the coastal zone of the Great Barrier Reef (GBR) (*McKinnon et al., 2013*), despite being located in the wet Australian tropics with a distinct rainy season. Moreover, the inshore area was even more strongly autotrophic than the offshore region of the GBR, which is in contrast to the expectation that inshore coastal waters should be heterotrophic. Coastal waters adjacent to Northern Australia were also predominantly autotrophic (*McKinnon et al., 2017*). Whether autotrophic metabolism could prevalent in other regions of Australia is, thus far, unresolved.

Here we report plankton metabolic rates for two contrasting coastal sites in the Perth area in temperate Western Australia, Matilda Bay, an inshore-site in the Swan river estuary, and Woodsman Point, an open coastal site. Specifically, we assessed biweekly during a year (March 2014 to March 2015) community respiration, gross primary production and net community production, along with temperature, salinity, dissolved inorganic nutrient concentration, and chlorophyll *a* concentration.

## METHODS

Matilda Bay is located in the lower reaches of the Swan River. Swan is one of the main rivers in Western Australia with an extension of more than 50 km and a catchment area about 190,000 $Km^2$ (*Thompson, 1998*). The estuary has been open to the ocean since 1987

when a rocky bar near to the mouth of the estuary in Fremantle was removed. The estuary has a seasonal cycle influenced by rainfall with wet and cool winters with about 90% of the annual rain and hot and dry summers (*Thompson, 1998*; *Hamilton et al., 2006*). During winter most of the water body is fresh because of the rainfall and runoff but the salinity increases upstream when the rainfall decreases and the system receive a significant flow of oceanic waters (*Thompson, 1998*). The estuary has received anthropogenic pressure because of land clearing for agricultural purposes, urbanization, dam construction and other factors (*Chan et al., 2002*; *Thompson, 1998*). Consequently, nutrient inputs and sedimentation rates have increased and the water quality has decreased (*Chan et al., 2002*; *Hamilton et al., 2006*). *Gedaria (2012)* reported that salinity and temperature are the main drivers of the variation in phytoplankton biomass in the Swan River estuary. Woodman Point is located in the Owen Anchorage in the Coast of Cockburn Sound (Perth, Western Australia), and, in contrast with Matilda Bay, represents an open shoreline with no direct freshwater influence. Cockburn Sound area was a place of industrial activity reducing coastal water quality in the 1950 s due to phytoplankton blooms (*Cambridge & McComb, 1984*). However, the water quality has improved since the mid-1990 s due to the reduction of nutrients inputs from industrial activities (*Kendrick et al., 2002*). The coastal area of Western Australia is also influenced by the Leeuwin Current, a poleward-flowing eastern boundary current characterized by warm waters with low salinity and low nutrients, but weakening between November and March (*Cresswell & Golding, 1980*).

Sub-surface water samples were collected at biweekly intervals between March 2014 to March 2015 in Matilda Bay (−31.9904°S, 115.8181°E; Fig. 1) and the Ammunition Jetty, Woodman Point (−32.1241°S, 115.7586°E; Fig. 1). The free software Ocean Data View (*Schlitzer, 2016*), version 4.7.10, was used to generate the study area map. Samples were then transported to incubated and processed at the University of Western Australia (UWA). Temperature (°C), salinity and dissolved oxygen were measured through the water column by deploying a calibrated YSI EXO1 Multi-parameter Water Quality Sonde fitted with a pressure sensor (±0.04 m), temperature (±0.01 °C), conductivity sensor, and an optode dissolved oxygen sensor. In addition, surface water temperature was measured from the water collected by a digital thermometer.

Net community production, gross primary production, and community respiration were quantified by changes in dissolved oxygen using micro-Winkler techniques by the use of a precise automatic titration based on redox potentiometric endpoint (*Oudot et al., 1988*). Water collected in each site was siphoned into 21 calibrated glass 100 ml borosilicate Winkler bottles. Seven bottles were fixed immediately to measure initial oxygen, another seven bottles were incubated in the light and the last seven bottles were incubated in the dark. The incubation, starting between 8 to 9 AM, was run for 24 h *in situ* conditions of temperature and natural solar radiation in an outdoor, temperature controlled tank. After the incubation, samples were fixed and the final oxygen was measured using a high-precision autotitrator (Compact Titrator G20; Mettler Toledo, Columbus, OH, USA). NCP rates were determined from the oxygen change in the clear bottles (oxygen clear −initial oxygen), CR rates were determined from the oxygen change in the dark bottles (initial oxygen–dark oxygen) and GPP rates were calculated as the sum of CR and NCP

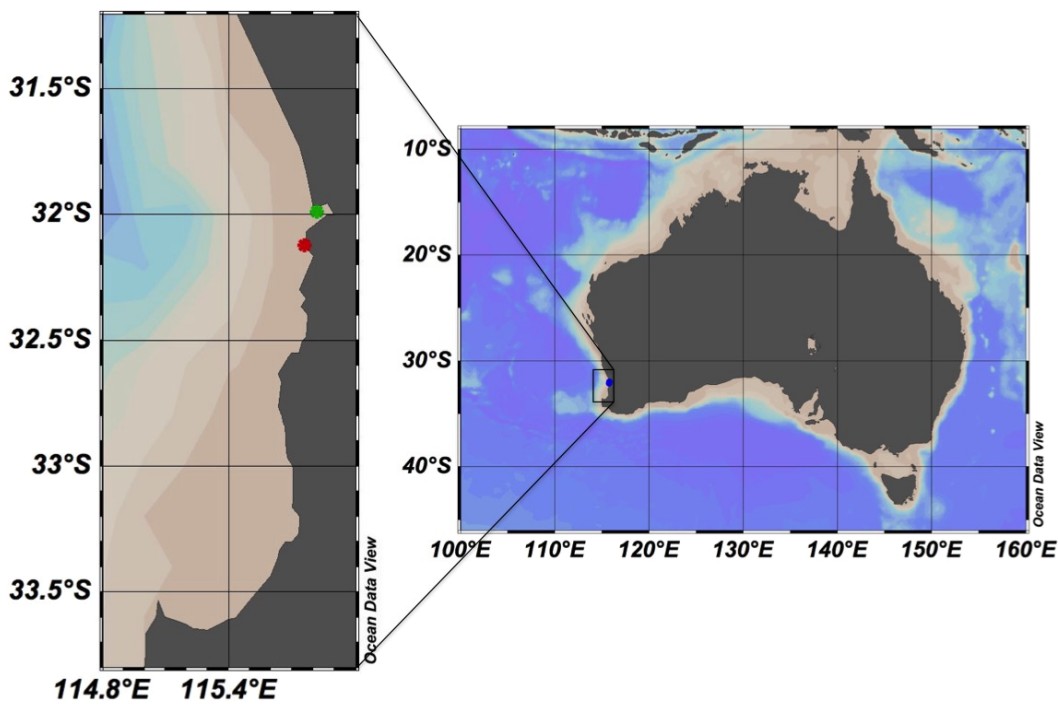

**Figure 1 Map showing the study area in Western Australia.** The stations sampled in the coast of Perth are marked with green (Matilda Bay) and red (Woodman Point) dots. The free software Ocean Data View (*Schlitzer, 2016*), version 4.7.10, was used to generate the study area map.

(*Duarte, Agustí & Regaudie-de Gioux, 2011*). Model II regression was used to analyse the relationships between CR, NCP, and GPP.

Aliquots of 200 ml were used for chlorophyll *a* analyses using acetone extraction and fluorometric determination after *Parsons, Maita & Lalli (1984)*. Subsamples were filtered through Whatman grade GF/F glass microfiber filters of 25 mm diameter. Filters were placed in plastic tubes of 1 ml and stored at −20 °C until analysis. Filters were immersed in acetone at 90% during 24 h for chlorophyll *a* extraction. After that period, chlorophyll *a* fluorescence was measured by the use of a Trilogy Laboratory Fluorometer (Turner Designs) equipped with a module of Chlorophyll *a* Non-Acidification Fluorescent Module (CHL-A NA) at UWA. The fluorometer was calibrated with pure chlorophyll *a* (Sigma-Aldrich C6144-1mg; Sigma-Aldrich, St. Louis, MO, USA) solution.

Samples for dissolved inorganic nutrient analyses were collected during transportation to the laboratory and kept frozen until analysis in a segmented flow autoanalyzer following standard procedures Samples (*Hansen & Koroleff, 1999*).

The temperature response of plankton communities was described by fitting, using least squares regression analysis, the Arrhenius equation,

$$\text{Ln } Y = A \exp^{-AE/kT}$$

**Table 1** Mean (± SE) of the variables measured in Matilda Bay and Woodman Point (Western Australia).

| | Matilda bay | | Woodman point | |
|---|---|---|---|---|
| | **Mean** | **± SE ($N = 33$)** | **Mean** | **± SE ($N = 27$)** |
| Temperature (°C) | 20.90 | 0.66 | 20.79 | 0.57 |
| Salinity | 32.09 | 0.95 | 35.13[*] | 0.29 |
| Ammonia ($\mu$mol N L$^{-1}$) | 2.68 | 0.30 | 2.17 | 0.29 |
| Nitrate ($\mu$mol N L$^{-1}$) | 2.32 | 0.71 | 0.56[*] | 0.08 |
| Phosphate ($\mu$mol P L$^{-1}$) | 0.52 | 0.06 | 0.20[*] | 0.02 |
| Chlorophyll a ($\mu$g Chl $a$ L$^{-1}$) | 4.05 | 0.47 | 1.68[*] | 0.14 |
| Respiration ($\mu$mol O$_2$ L$^{-1}$ d$^{-1}$) | 8.81 | 0.76 | 4.33[*] | 0.48 |
| NCP ($\mu$mol O$_2$ L$^{-1}$ d$^{-1}$) | 7.21 | 1.29 | 2.32[*] | 0.61 |
| GPP ($\mu$mol O$_2$ L$^{-1}$ d$^{-1}$) | 16.05 | 1.56 | 6.23[*] | 0.60 |
| P/R | 1.91 | 0.14 | 2.05 | 0.27 |

Notes.

NCP, net community production; GPP, gross primary production; P/R, is the ratio of GPP over R.

[*]denotes statistically significant difference ($t$-test, $P < 0.05$).

where $Y$ is the property of interest, AE is the activation energy (eV), $k$ is the Boltzmann's constant (8.617734 10–5 eV°K$^{-1}$) and T is the sea-surface water temperature (°K), and A is a fitted intercept (*Regaudie-de Gioux & Duarte, 2012*).

# RESULTS

Surface water temperature ranged from 12.0 to 27.4 °C and 15.1 to 25.0 °C (Fig. 2A) and salinity ranged from 22.03 to 36.97 PSU and 31.6 to 37.1 PSU (Fig. 2B) in Matilda Bay and Woodman Point, respectively. The minimum salinity was reached in late winter and spring in Matilda Bay, following river discharge, and while the pattern was less clear in Woodman Point, the lowest salinity was also observed in winter and early spring (Fig. 2B). Dissolved inorganic nitrogen concentration was highest in winter, but phosphate concentration was highest in late summer in Matilda Bay (Figs. 3A–3C ). In contrast, nitrate and phosphate concentrations in Woodman Point were lower ($P < 0.05$) than those in Matilda Bay (Table 1) and dissolved inorganic nitrogen concentration showed two maxima, winter and summer, while phosphate concentrations showed a summer minima (Figs. 3A–3C). Chlorophyll *a* concentration was significantly higher and more variable in Matilda Bay than in Woodman Point (Table 1, Fig. 3D), and reached the highest values in winter, at the time of peak nitrate concentration (Fig. 3D), as there was a significant, positive, relationship between chlorophyll *a* concentration and nitrate concentration ($r = 0.67$, $P < 0.0001$).

Respiration rates were, on average, twice as high in Matilda Bay as in Woodman Point (Table 1), and increased strongly toward summer in Woodman Point whereas it showed less seasonal variability in Matilda Bay (Fig. 4A). Gross primary production was also much higher in Matilda Bay than in Woodman Point (Table 1), with no clear seasonal pattern at either site (Fig. 4B). The communities were generally autotrophic, with GPP about twice as high as R (NCP and P/R >1, Table 1), with NCP being three times higher, on average, at
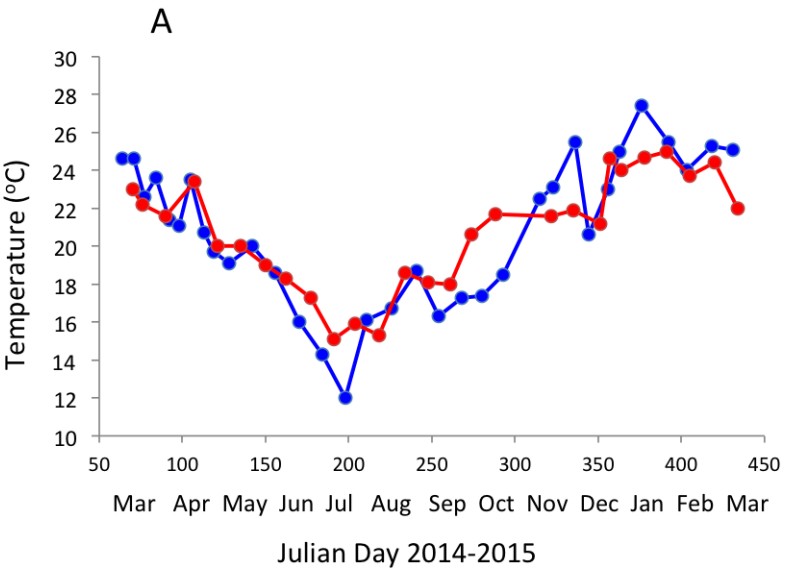

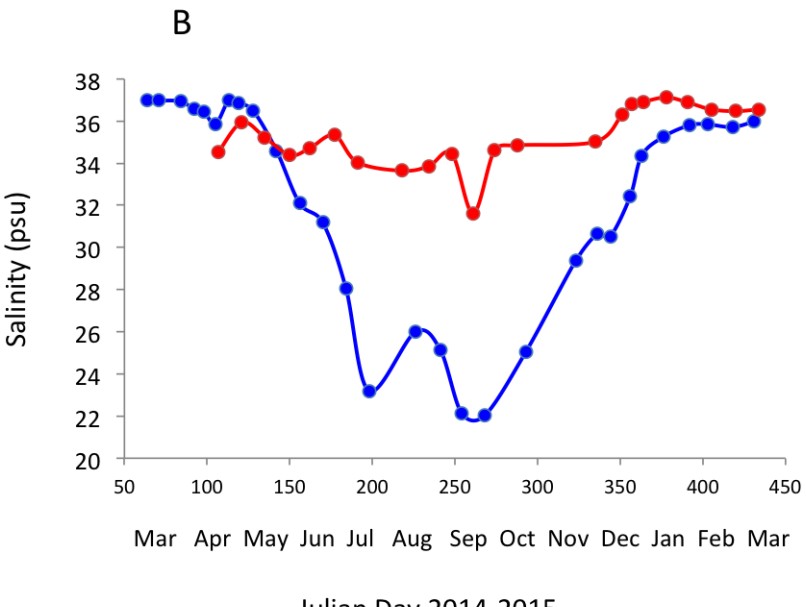

**Figure 2 Temporal variability in seawater temperature and salinity.** Changes in surface seawater temperature (A) and salinity (B) in Matilda Bay are shown in blue line and symbols, and in Woodman Point in red line and symbols, over time.

Matilda Bay than at Woodman Point (Table 1, Fig. 4C), and neither community displayed any clear seasonal trend in net community production along the year (Fig. 4C). GPP was significantly correlated with CR ($r = 0.68$, $P < 0.0001$), but NCP increased strongly with increasing GPP (Fig. 5). Net community production and gross primary production increased with increasing chlorophyll $a$ concentration (NCP, $R^2 = 0.73$, $P < 0.0001$;

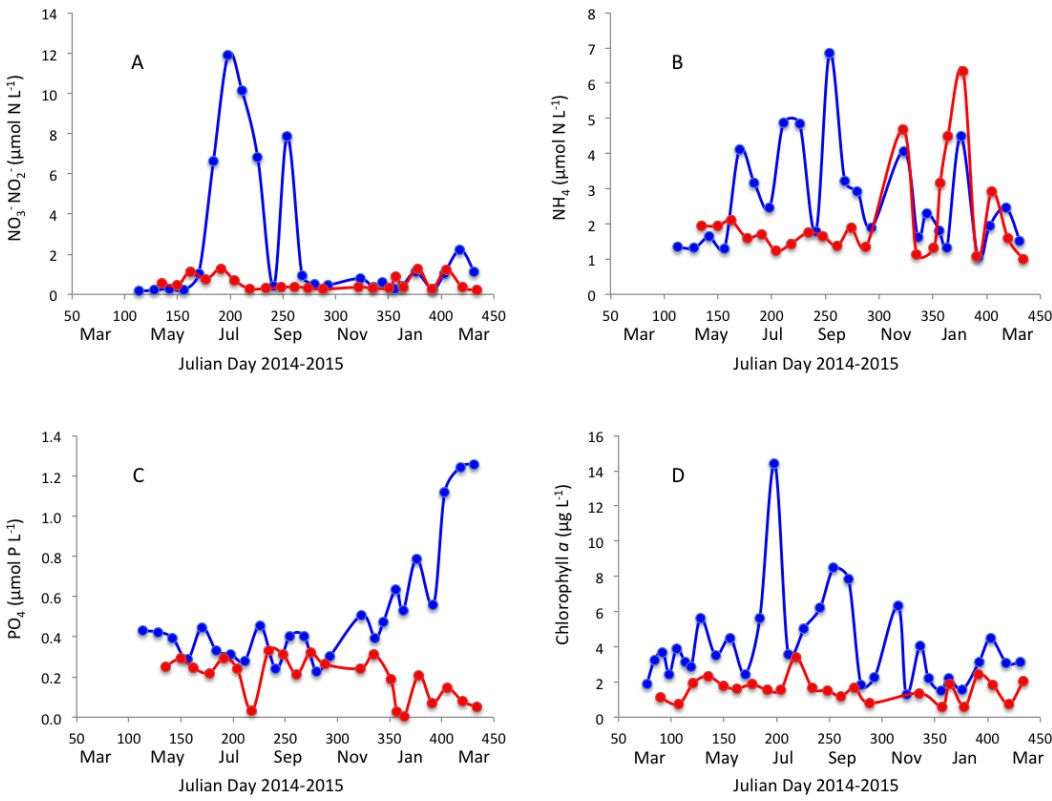

**Figure 3** **Nutrients and chlorophyll *a* variability.** Changes in (A) nitrate, (B) ammonium, (C) phosphate and (D) chlorophyll *a* concentration in Matilda Bay (blue line and symbols) and Woodman Point (red line and symbols) over time.

GPP, $R^2 = 0.69$, $P < 0.0001$), with the relationship between community respiration rate and chlorophyll *a* being much weaker (CR, $R^2 = 0.15$, $P = 0.0036$) albeit also significant (Figs. 6A–6C).

Respiration rates increased with increasing temperature, resulting in an activation energy of $0.76 \pm 0.21$ eV (Fig. 7A). GPP showed, in contrast, no significant temperature-dependence ($P > 0.05$). Indeed, when standardized to chlorophyll *a*, as observed in previous studies (e.g., *Regaudie-de Gioux & Duarte, 2012*; *Garcia-Corral et al., 2017*), there was a significant temperature-dependence of gross primary production, with an activation energy of $0.69 \pm 0.12$ (Fig. 7B), comparable to that of community respiration.

## DISCUSSION

Chlorophyll *a* values were higher in Matilda Bay than in Woodman point, but all values ranged within those reported for coastal waters around Perth (*Pearce, Lynch & Hanson, 2006*). The highest chlorophyll *a* concentration in both Matilda Bay and Woodman point occurred at the low salinity winter events. Chlorophyll *a* concentration in the Swan River has been reported to vary seasonally showing large interannual variability (*Thompson, 1998*).

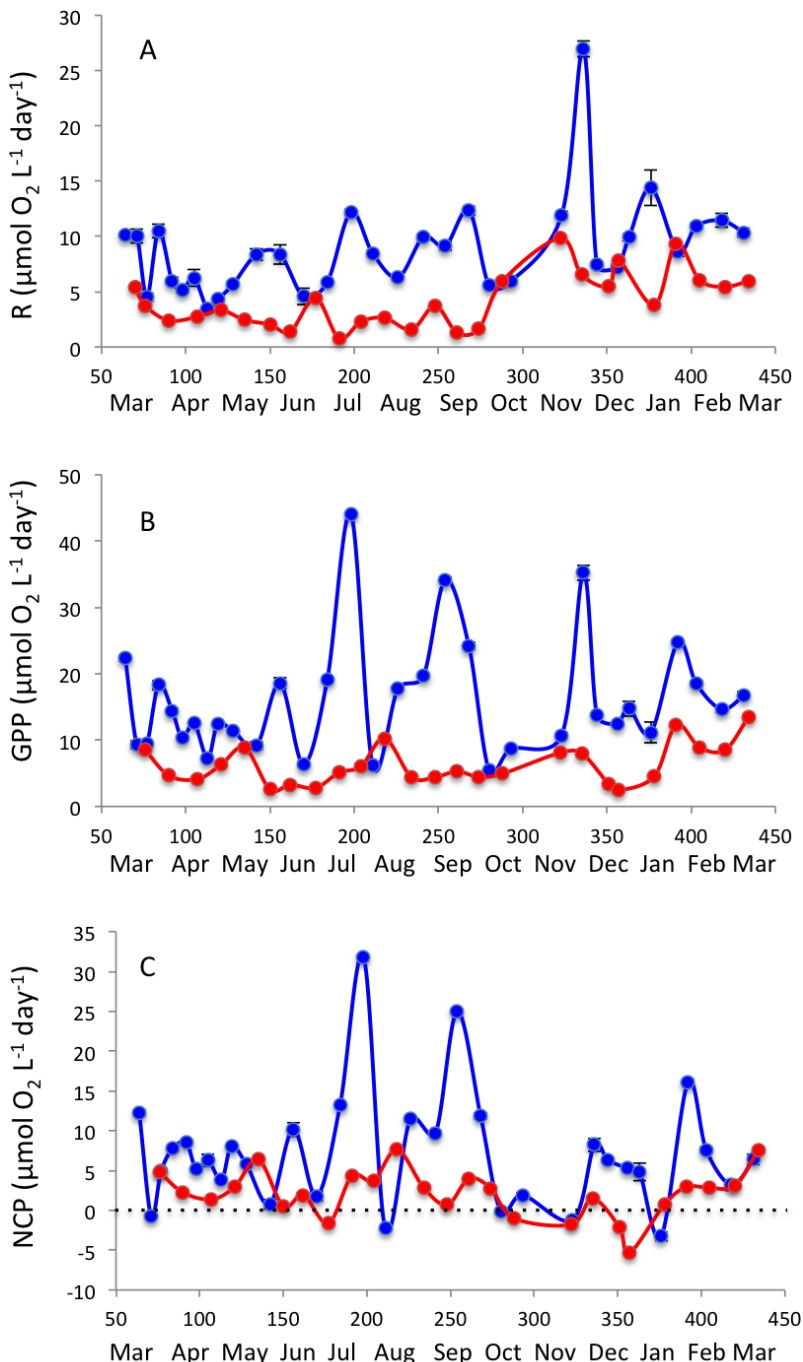

**Figure 4** **Planktonic metabolism.** Changes in (A) community respiration rate, (B) gross primary production, and (C) net community production in Matilda Bay (blue line and symbols) and Woodman Point (red line and symbols) over time.

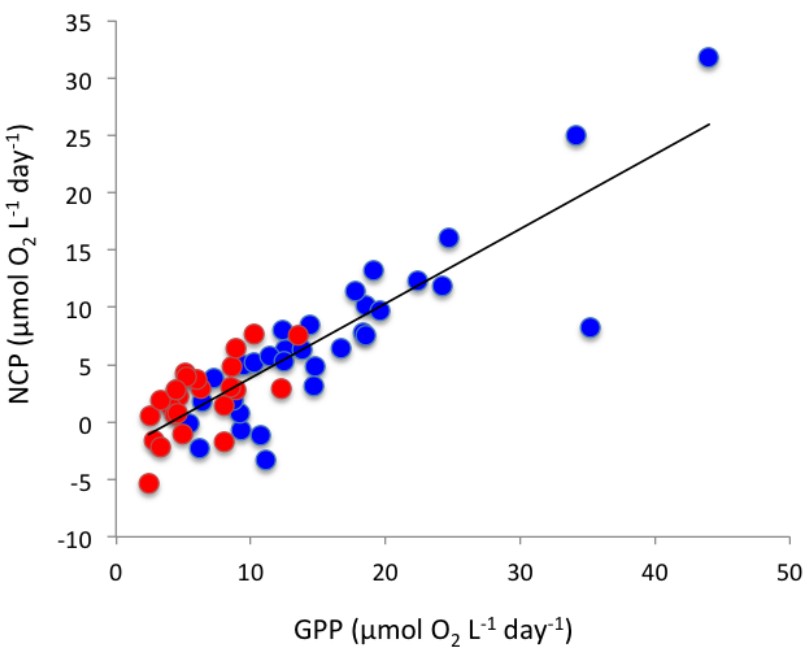

**Figure 5** **The relationship between net community production and gross primary production** The solid line shows the fitted regression equation: NCP (mmol $O_2$ m$^{-3}$ d$^{-1}$) $= -2.65 + 0.65(\pm 0.05)$ GPP (mmol $O_2$ m$^{-3}$ d$^{-1}$) ($R^2 = 0.75$, $P < 0.0001$). Blue symbols and red symbols correspond to Matilda Bay and Woodman Point, respectively.

Both coastal ecosystems, but particularly Matilda Bay, supported productive communities, as reflected in high GPP rates.

Community respiration rates were less variable than GPP, particularly at Matilda Bay, but GPP sufficed to support all carbon demands from the community generating excess organic matter. This resulted in the prevalence of autotrophic communities at both sites, with average P/R ratios above 2.0 similar across both sites. This is expected from relatively productive sites with GPP well above the threshold previously determined to delineate autotrophic from heterotrophic communities (*Duarte & Agustí, 1998*; *Duarte & Regaudie-de Gioux, 2009*). Net community production was strongly correlated with chlorophyll *a* concentration, accounting for the much higher NCP in productive Matilda Bay compared to Woodman Point plankton communities, suggesting that the metabolic balance of plankton communities in the coast of Perth is regulated by factors controlling autotrophic processes, such as nutrient inputs, salinity regimes and temperature. In coastal waters of Northern Australia, *McKinnon et al. (2017)* also observed that the metabolism and community respiration were autotrophic and positively related to chlorophyll *a* concentration.

The results presented here contribute to address a paucity of studies of plankton community metabolism in the Indian Ocean (*Regaudie-de Gioux & Duarte, 2013*). *Robinson & Williams (1999)* studied the planktonic metabolic balance during a research cruise in the Gulf of Oman, reporting P/R ratios for surface waters between 1.17 and 2.43, with

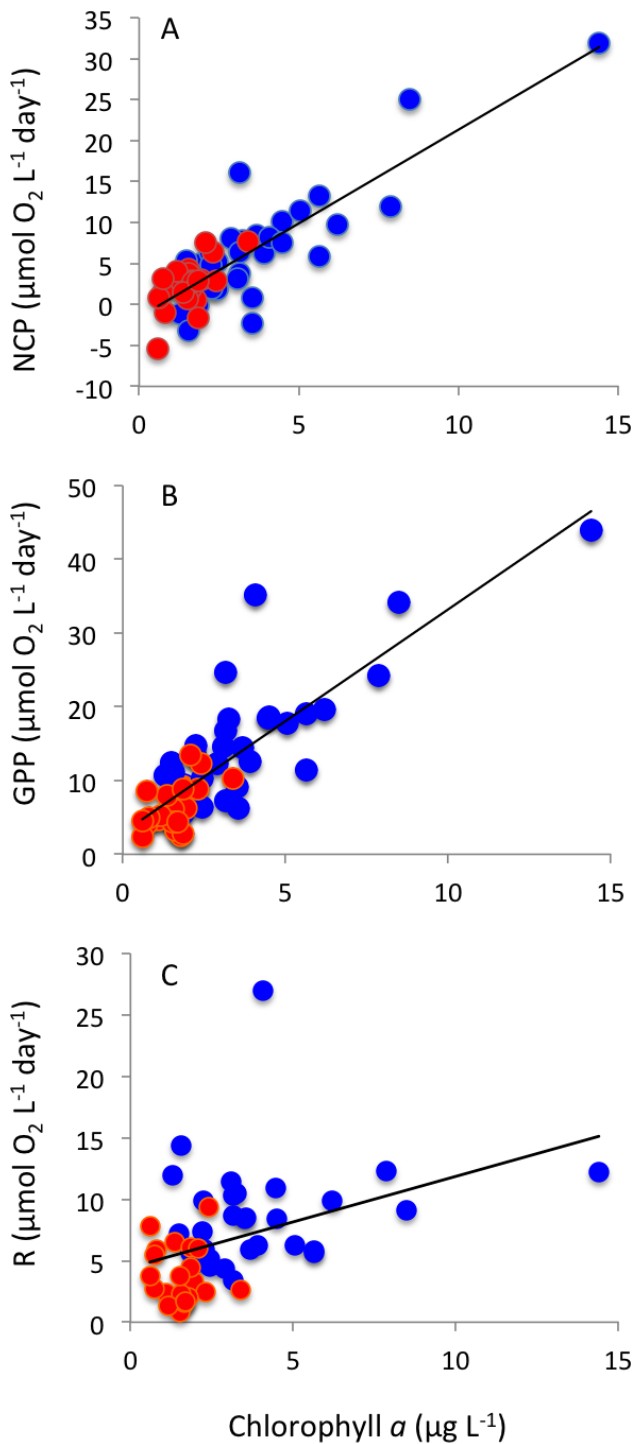

**Figure 6** **Plankton metabolism and phytoplankton.** The relationship between (A) net community production, (B) gross primary production and (C) community respiration and chlorophyll $a$ concentration. The solid lines show the fitted regression equations: (A) NCP (mmol $O_2$ m$^{-3}$ d$^{-1}$) $= -1.54 + 2.29$ ($\pm 0.19$) Chl $a$ ($\mu$g Chl $a$ L$^{-1}$) ($R^2 = 0.73$, $P < 0.0001$); (B) GPP (mmol $O_2$ m$^{-3}$ d$^{-1}$) $= 2.93 + 3.03$ ($\pm 0.05$) Chl $a$ ($\mu$g Chl $a$ L$^{-1}$) ($R^2 = 0.69$, $P < 0.0001$); and (C) R (mmol $O_2$ m$^{-3}$ d$^{-1}$) $= 4.45 + 0.74$ ($\pm 0.24$) Chl $a$ ($\mu$g Chl $a$ L$^{-1}$) ($R^2 = 0.15$, $P = 0.0036$). Blue symbols and red symbols correspond to Matilda Bay and Woodman Point, respectively.

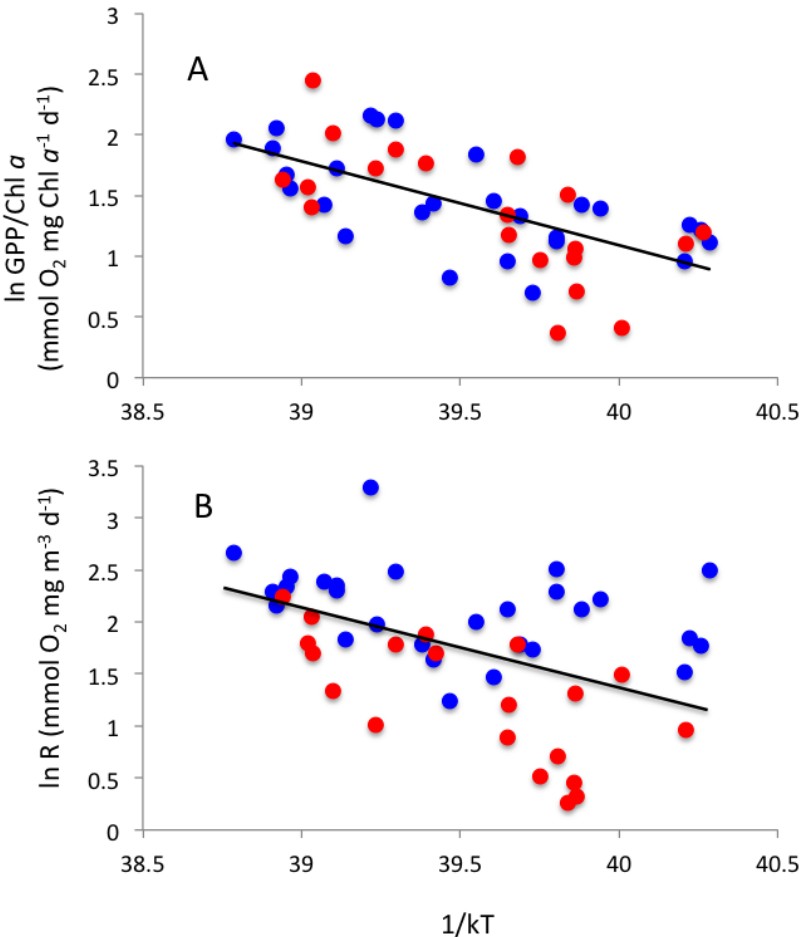

**Figure 7 Thermal relationships.** Arrhenius plots showing the relationship between the natural log of (A) community respiration and (B) gross primary production standardized to chlorophyll $a$, and $1/kT$, where k is the Boltzmann's constant ($8.617734\ 10^{-5}$ eV °K$^{-1}$) and T is the sea-surface water temperature (°K) in Matilda Bay (blue symbols) and Woodman Point (red symbols). The solid lines show the fitted equations: ln R (mmol $O_2$ mg m$^{-3}$ d$^{-1}$) = 31.87–0.76 ($\pm0.21$) $1/kT$ ($R^2 = 0.21$, $P = 0.0008$) and ln GPP/Chl a (mmol $O_2$ mg Chl $a^{-1}$ d$^{-1}$) = 28.8–0.69 ($\pm0.12$) $1/kT$ ($R^2 = 0.41$, $P < 0.0001$).

the highest ratio near to the Omani coast (*Regaudie-de Gioux & Duarte, 2013*; *Robinson & Williams, 1999*). Indeed, the P/R ratio of the station closer to the Omani coast, 2.43 (*Robinson & Williams, 1999*) was similar to the P/R ratio for out study sites in the Western Australia coast. More recently, *McKinnon et al. (2017)* in a comprehensive study of the metabolism of planktonic communities in the North of Australia, found that Indian Ocean communities from Ningaloo Reef and Exmouth Gulf (North Western Australia coast) were also predominantly net autotrophic.

Our results indeed showed a prevalence of net autotrophic metabolism in plankton communities of the Coast of Perth (Western Australia), both at the productive estuarine waters at Matilda Bay and the open coastal Indian Ocean waters at Woodman Point. This indicates that planktonic communities in these coastal locations act as strong $CO_2$

sinks and sources of organic matter and oxygen to the system. This is in contrast to the expectation that near-shore ecosystems act as sources of atmospheric $CO_2$, proposed to reconcile opposing views on the role of coastal ecosystems as $CO_2$ sources or sinks (*Chen & Borges, 2009*; *Cai, 2011*). Indeed, the pattern showed here, with higher net community production in the inner waters of Matilda Bay compared to the more open waters at Woodman Point agree with prior findings for Australian GBR where inshore areas being more strongly autotrophic than offshore waters (*McKinnon et al., 2013*).

The plankton communities in the coastal waters of Perth showed increased metabolic rates with increasing temperature, as expected from metabolic theory of ecology (*Brown et al., 2004*). However, the activation energy for gross primary production of 0.69 ± 0.12 eV found here was well below that found in previous analyses of Indian Ocean communities. *Garcia-Corral et al. (2017)* reported for Indian Ocean open-ocean waters a gross primary production Ea (standardized to chlorophyll *a*) of 1.70 eV. Also, whereas consistent with other assessments (e.g., *Regaudie-de Gioux & Duarte, 2012*; *Garcia-Corral et al., 2017*), the activation energy for community respiration was higher than that for gross primary production, this difference was small and not statistically significant. This is important as it predicts that warming events, such as the heat wave that impacted marine ecosystems across Western Australia in 2011 (*Wernberg et al., 2016*), will affect gross primary production and respiration rates of plankton communities in a similar way.

The net autotrophic communities encountered in the coastal system studied in Western Australia suggest that these pelagic communities produce organic matter in excess, thereby exporting organic matter either to the underlying benthic compartment or offshore. For instance, analysis of carbon stocks in Matilda Bay sediments, which support seagrass meadows, have shown the organic carbon to be partially of planktonic origin along with contributions of the seagrass themselves (*Rozaimi, Serrano & Lavery, 2013*). Hence, both the pelagic and benthic compartments of Matilda Bay appear to be autotrophic, although the contributions of the benthic compartment to ecosystem metabolism were not addressed here. We, therefore, provide only one of the components, pelagic metabolism, that determine ecosystem metabolic budgets, which include contributions from benthic compartments as well as exchanges, either import or export, with adjacent ecosystems.

## CONCLUSIONS

Our results indicated that planktonic communities in the two coastal Western Australia locations studied act as strong $CO_2$ sinks and sources of organic matter and oxygen to the system. The plankton communities of the Coast of Perth (Western Australia) showed net autotrophic metabolism both at the productive estuarine waters at Matilda Bay and the open coastal Indian Ocean waters at Woodman Point. This result is in contrast to the expectation of net heterotrophic balance for near-shore ecosystems, but in agreement with the few metabolic balance assessments from Australian coastal waters. The thermal relationships indicated that warming may decrease the strong capacity observed for $CO_2$ sinks. Our study is based on two contrasting plankton communities in Western Australia and, while useful to address the absence of reports on plankton community metabolism

in the Indian Ocean coast of Australia, and the paucity of reports across the Indian Ocean (*Regaudie-de Gioux & Duarte, 2013*), a broader analysis of coastal plankton communities across Western Australia is required to confirm the patterns revealed here and diagnose the role of plankton communities in across Western Australia in carbon fluxes and their likely response to future warming.

## ACKNOWLEDGEMENTS

We thank Lara García Corral and María Comesaña for their assistance with sampling and analyses.

### Funding

This research was funded by the Australian Research Council's Discovery Project (project number DP140100825) and the King Abdullah University of Science and Technology (KAUST) base line funding BAS/1/1072-01-01 to Susana Agusti. The funders had no role in study design, data collection and analysis, decision to publish, or preparation of the manuscript.

### Grant Disclosures

The following grant information was disclosed by the authors:
Australian Research Council's Discovery Project: DP140100825.
King Abdullah University of Science and Technology (KAUST): BAS/1/1072-01-01.

### Competing Interests

Susana Agusti is an Academic Editor for PeerJ. Lorena Vigoya is employed by AECOM. She participated in the study while getting her masters degree at the University of Western Australia.

### Author Contributions

- Susana Agusti conceived and designed the experiments, analyzed the data, contributed reagents/materials/analysis tools, prepared figures and/or tables, authored or reviewed drafts of the paper, approved the final draft.
- Lorena Vigoya performed the experiments, analyzed the data, authored or reviewed drafts of the paper, approved the final draft.
- Carlos Manuel Duarte conceived and designed the experiments, analyzed the data, contributed reagents/materials/analysis tools, authored or reviewed drafts of the paper, approved the final draft.

### Data Availability

    The raw data are provided in Data S1.

## Supplemental Information

Supplemental information for this article can be found online at http://dx.doi.org/10.7717/peerj.5081#supplemental-information.

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
