# Peer review of "Annual plankton community metabolism in estuarine and coastal waters in Perth (Western Australia)"

_PeerJ, doi:10.7717/peerj.5081_

## Round 0.1 · original submission · Minor Revisions

· Academic Editor

Minor Revisions

All three reviewers recommended your manuscript be published after minor revisions, please see their specific comments below. In particular, all three pointed out the need to correct grammatical errors and to restructure/shorten sentences. They also provided suggestions to improve the discussion. Please provide a detailed point-by-point reply to each of the reviewers' comments.

I look forward to receiving the revised manuscript.

·

Basic reporting

The quality of the written English, though adequate, could be improved by breaking some of the more complex sentences into simpler, shorter sentences.

Experimental design

The experimental design is sufficient to address the goals of the study, and has the great strength of regular fortnightly measurements over a year.

Validity of the findings

As the authors note, this is the only study of coastal pelagic metabolism in temperate Australia. The study is well conducted and the methods and conclusions appropriate.

Additional comments

Line 68: Both study sites are within the greater Perth metropolitan area (Perth has a population of over 2 million).
Line 70: The study by McKinnon (2017) includes a number of Australian coastal areas, not just the Great Barrier Reef study of McKinnon (2013). Your point that autotrophic plankton metabolism prevails is still valid - but applies in a number of sites. The main contrast between your study and those included in McKinnon (2017) is that your study is the only one in temperate Australia. You could probably build on on this contrast a little more than you have.
Line 84: you mention bacterial abundance here, but present no data and no analyses of bacterial abundance as a predictor of metabolic processes, other than a comment in the abstract that no correlations were observed.
Line 118: I note the authors used borosilicate bottles, a practice they have criticised elsewhere on the basis of these not being UV transparent.
Lines 213-219: Note other Indian Ocean sites described in McKinnon et al (2017).
Line 222: I suggest you remove the word eutrophic - which these days is a loaded term. Maybe "productive" might be better.
Line 231: Not just GBR, as above.
Line 243: If GPP and CR will be affected in a similar way, what ecosystem effects do you predict?
Figures 4&5: I assume you used Model II regression?

·

Basic reporting

The authors clearly reported their findings: their study was outlined well and the results were easy to follow. There are some grammatical issues that I commented on that should be addressed. Additionally, several of the sentences are very comma heavy. While most of the instances are grammatically correct, it gets hard to follow. I would recommend trying to reduce comma use by restructuring some of the sentences.
Their review of the literature was thorough. It may be possible to contextualize what may be happening to the organic matter that is being produced more than what they have done.
The whole piece (article, figures, tables, and data) was well done.
The work was observatory in nature and the authors clearly achieved this goal.

Experimental design

The experimental design was impressive in terms of scope—they made regular measurements for a year and fill existing gaps in the literature (very little plankton metabolism work has been done for coastal waters in that area).
The research was more observatory in nature and they fulfilled this aim well.
The methods were described very well. The only comments I have for this section is to mention at what time their incubations were started/finished and to also give a little more background information about Woodman Point.

Validity of the findings

I think the authors contextualized their findings well. I think it would have been possible to discuss what they think may be happening to the organic matter that is being produced more than what they have done.
The data was robust and their conclusions are well stated.

Additional comments

This work assessed community respiration, net community production, and gross primary production at two locations off the West coast of Australia.
The results the authors present are impressive in terms of scope—they made regular measurements for a year and fill existing gaps in the literature (very little plankton metabolism work has been done for coastal waters in that area).
I think that the journal and oceanographic literature would benefit from this work being published. The biggest benefit would be that future metabolism work in the Southern hemisphere and the Indian Ocean will have now have existing work to contextualize their own research and modelers will have more data to build from.
There are a few minor issues that I think need to be addressed in order to ensure the work is as clear to readers as possible. I have added line by line comments that may help to improve the work (mostly grammatical in nature). Additionally, I think the end of the discussion would be improved by the authors mentioning what they think might be happening to the organic matter that is being locally produced. Are there any estimates about the rate of sedimentation in the area? Do they think the currents are taking it elsewhere where it is being respired? Etc.

Introduction
Line 54: Use the abbreviations GPP and R here (or CR if you choose CR rather than R throughout the paper).
Line 57-59: These two sentences seem related enough that they could be one sentence.
Line 66-70: Linking these sentences to the previous paragraph might help to guide the reader a bit. Something along the line of: While this has important implications, virtually all of the above results are from work in highly populated regions of the Norther hemisphere (i.e. Europe, the USA, and Asia) and plankton metabolism may differ in Southern hemisphere waters.
Line 75: You mentioned the Great Barrier Reef above but did not define GBR to be the abbreviation.
Line 81: I think the word fortnightly might confuse some readers. That is just a personal preference though.
Line 82: I think it is confusing that you refer to community respiration as R earlier in the introduction and as CR here. It seems like you prefer CR so I would stick with that. You also don’t have to redefine the abbreviations for GPP and CR since they were defined above.

Methods
Line 90: Change: “The estuary is permanently…” to “The estuary has been permanently…”
Line 92-93: This seems oddly phrased to me. It might be improved by slightly different punctuation or reordering of the sentence.
Line 97: Change to “purposes, urbanization, dam construction, and other factors…”
Lines 88-103: You spend so much time orienting us to Matilda Bay but so little on Woodman Point. It would be good to orient the reader to Woodman Point a little more (how many kilometers off shore, is there seasonal upwelling, etc.)
Line 105: Again, I would suggest a different word choice for the frequency at which you took your samples.
Line 115: You don’t have to redefine the abbreviations here.
Line 120: What time of day did you start incubations?
Line 128: Delete “of water samples”
Line 134: Italicize the “a” after chlorophyll.

Results:
Line 153: Replace “units” with “PSU”
Line 155-156: This is oddly phrased. I would suggest: “and while the pattern was less clear in Woodman Point, the lowest salinity was also observed in water and early spring.”
Line 158: Write phosphate instead of phosphorus.
Line 160. Delete “in”
Line 168: Shows should be past tense and delete “a”
Line 172: Change “that” to “than”
Line 173: “Displaying” should be past tense.
Lines 175-179: Do you have stats for these statements? P values? R2 values?
Line 182: This interpretation seems better suited for the discussion.

Discussion
Line 190: Change “which” to “but all” and cite both Pearce and Thompson in Line 191.
Line 194-196: I think this sentence should be deleted because you already spoke about Chl a values being in range above.
Line 199: “was” should be “were” or you should delete the word “rates”; also, you shouldn’t capitalize “Respiration”
Line 235-238: There are a lot of commas in this sentence and I can’t follow the train of thought. I noticed that high commas use was an issue throughout the paper. While it is grammatically correct in most instances it makes many of the sentences hard to follow. I think it would be possible to reduce the use of commas by restructuring sentences in many cases.
Line 242: Delete “on”

Figure 1 caption: Change “as blue line and symbols” to “in blue” and change “as red line and symbols” to “in red” and change “, along the study period” to “over time”
Figure 2 and 3 captions: Change “along the study period” to “over time”
Figure 5 caption: Change “shows” to “show”

·

Basic reporting

The authors do a good job setting up the context for this study, including the importance of understanding the role of coastal metabolism in global carbon budgets. Thus, measures of NCP and CR are useful for placing the coastal margins into a broader context. The figures are generally well-drafted, showing temporal patterns in the key results, and relationships between metabolism and key environmental variables (temperature, chlorophyll a). My biggest gripe is that there are numerous grammatical issues, including verb-tense disagreement, missing articles, and wordiness. The manuscript would be significantly improved with a careful edit to improve grammar and to reduce redundancy. I also think a map showing the location of sampling sites would be helpful to readers unfamiliar with this area.

Experimental design

The authors point out that measurements of plankton metabolism are rare in coastal environments of the southern hemisphere, thus questioning whether generalities drawn from the existing literature studies are sound given that they are predominately drawn from studies in the northern hemisphere. The sampling design is straightforward and clear, repeatedly sampling two sites over an annual cycle and conducting metabolism experiments on collected water. The experimental design of the metabolism experiments is also straight-forward and the methods appear appropriate for dissolved oxygen, chlorophyll, and nutrients. One omission or inconsistency I note was that the authors refer to bacteria samples, but no such data were presented.

Validity of the findings

The main findings from this paper were that the sites studied consistently displayed autotrophic behavior, and that the estuarine site was more metabolically active than the open water site, which is consistent with the expectation that estuaries tend to have more nutrients and organic matter than more open water areas. I think the authors should acknowledge a limitation of their experimental design, given that they only considered processes in surface waters and this limits the scope of the results and interpretation. To address net ecosystem metabolism more holistically, one must grapple with processes occurring throughout the water column and at the sediment water interface, the latter being particularly critical in shallow environments (e.g., Kemp et al. 1992). I encourage the authors to speculate how their interpretations may be tempered by taking into account the likelihood that processes nearer the bottom will become more heterotrophic, due to light limitation. One pattern that I found intriguing was the strong peak in phosphorus in Matilda Bay during summer. Could the authors speculate as to the reason for this? Perhaps sediment P regeneration?

Kemp, W.M., Sampou, P.A., Garber, J., Tuttle, J., Boynton, W.R., 1992. Seasonal depletion of oxygen from bottom waters of Chesapeake Bay: roles of benthic and planktonic respiration and physical exchange processes. Marine Ecology Progress Series 85, 137-152.

Additional comments

I include here specific instances of grammatical errors that should be addressed, referenced by line number.

L25: Abstract states experiments were conducted from Mar. to Oct. 2014, but text and data suggests Mar 2014-Mar 2015.

L27: Extraneous use of the word ‘the’. Should read ‘…was net autotrohic…’.

L28: Replace ‘higher than one’ with ‘>1’.

L29 and L35-36: Repetitive statements about negative correlation with salinity.

L36-38: Broad statements about role of import/export of organic matter require a more comprehensive budget that includes contributions throughout the water column and the benthos.

L47: Replace ‘ration’ with ‘ratio’.

L47-48: Omit redundant phrase ‘, or experiences small deviations from this balance’.

L49: Replace ‘inputs’ with more generic term ‘fluxes’, given that inputs AND exports will affect P/R ratio.

L50: Omit ‘in the open ocean’ as this is already stated at the beginning of the sentence.

L51: Replace ‘present’ with ‘have’ (2 instances).

L58: Omit ‘What’. Start sentence with ‘The’. Also omit ‘is’.

L63: Misspelling: ‘fueled’.

L65: Add clause to complete the comparison. Something like: ‘, whereas outer reaches of estuaries tend to be CO2 sinks.

L89: Awkward sentence with two subjects: Matilda Bay and Swan River.

L89 and throughout: ‘Km’ should read ‘km’.

L90: Omit ‘permanently’, given that you go on to describe an alteration in 1987.

L92: Omit ‘rainfall seasonal variations with’

L100-101: Replace ‘abundance of phytoplankton species’ with something like ‘variation in phytoplankton biomass’

L106-107: Too many significant figures on latitude/longitude. Suggest that 4 decimal places are sufficient. Also, consider omitting words ‘latitude’ and ‘longitude’ as it is self-evident from the °S and °E notation.

L111: Omit ‘as well as’. Reword as: ‘temperature (…), conductivity, and an optode….’

L116: What does ‘micro-Winklers’ mean here?

L118: Volume of bottles?

L131: Omit ‘their’.

L152: Be consistent with significant figures: ‘12’ and ‘25’ should read ’12.0’ and ’25.0’.

L153: One decimal place for salinity is sufficient.

L160: Remove extraneous ‘in’.

L162: What do you mean by ‘significantly’? Omit or explain with statistics.

L174: Describing statistical relationships between GPP and CR is problematic, because they are not independent (GPP = NCP + CR).

L180: Provide units for AE.

L197: Regarding the phrase ‘relatively high GPP rates’: relative to what?

L199: Do you mean ‘NCP’ here instead of ‘GPP’?

L199: ‘Respiration’ should read ‘respiration’.

L224: Should read ‘locations’, not ‘location’.

Figs. 1-3: Awkward month abbreviations. I suggest ‘Jan’,’Feb’,’Mar’,’Apr’,’May’,’Jun’,’Jul’,’Aug’,’Sep’,’Oct’,’Nov’,’Dec’.

Figure 6: Misspelled. Should be ‘Arrhenius’.

---

## Round 0.2 · Minor Revisions

· Academic Editor

Minor Revisions

Thank you for the revised version of your manuscript, and for satisfactorily answering most of the reviewers' comments. This new version represents a clear improvement over the first one. However, several of the reviewers' comments were not addressed, particularly regarding the discussion that wasn't significantly modified. I attach a shortened version of the "rebuttal letter" that only contains those comments that were not addressed. Please either incorporate them in a new version, or if you choose not to (which is OK) clearly explain why.

Moreover, some sentences are still very long and comma-heavy, for instance l. 43-48, l. 48-53, l. 169-173, l. 178-182, l. 245-248, l. 266-272. Please break these sentences in 2 or 3 sentences wherever possible. This would improve readability.

A few additional personal comments:
- l. 31: one "correlation" and one "correlations", please be consistent
- l. 81: double comma
- l. 89: km not Km
- l. 100: a period is missing.
- l. 102: I imagine it should be "Cockburn Sound" (extra period).
- l. 111-112: remove the minus signs since the latitude is already labelled "°S".
- l. 112-114: mention of Ocean Data View is not needed in an introduction, this sentence should be moved to the figure caption (as it seems to be already with the figure but not l. 384-385).
- l. 114: "transported to incubated and processed" - did you mean "transported to incubate and process" or maybe "transported, incubated and processed"?
- l. 115: salinity not Salinity
- l. 126: "between 8 to 9" should be "between 8 and 9"
- l. 239: should be "agrees" (or "the patterns").

---

## Round 0.3 · accepted · Accept

· Academic Editor

Accept

Thank you for answering all the reviewers' comments and for improving the discussion.

#